

# Ultraviolet-reflective film applied to windows reduces the likelihood of collisions for two species of songbird

John P. Swaddle[1,2], Lauren C. Emerson[2], Robin G. Thady[2] and Timothy J. Boycott[2]

[1] Institute for Integrative Conservation, William & Mary, Williamsburg, VA, USA
[2] Biology Department, William & Mary, Williamsburg, VA, USA

## ABSTRACT

Perhaps a billion birds die annually from colliding with residential and commercial windows. Therefore, there is a societal need to develop technologies that reduce window collisions by birds. Many current window films that are applied to the external surface of windows have human-visible patterns that are not esthetically preferable. BirdShades have developed a short wavelength (ultraviolet) reflective film that appears as a slight tint to the human eye but should be highly visible to many bird species that see in this spectral range. We performed flight tunnel tests of whether the BirdShades external window film reduced the likelihood that two species of song bird (zebra finch, *Taeniopygia guttata* and brown-headed cowbird, *Molothrus ater*) collide with windows during daylight. We paid particular attention to simulate the lighting conditions that birds will experience while flying during the day. Our results indicate a 75–90% reduction in the likelihood of collision with BirdShades-treated compared with control windows, in forced choice trials. In more ecologically relevant comparison between trials where all windows were either treated or control windows, the estimated reduction in probability of collision was 30–50%. Further, both bird species slow their flight by approximately 25% when approaching windows treated with the BirdShades film, thereby reducing the force of collisions if they were to happen. Therefore, we conclude that the BirdShades external window film will be effective in reducing the risk of and damage caused to populations and property by birds' collision with windows. As this ultraviolet-reflective film has no human-visible patterning to it, the product might be an esthetically more acceptable low cost solution to reducing bird-window collisions. Further, we call for testing of other mitigation technologies in lighting and ecological conditions that are more similar to what birds experience in real human-built environments and make suggestions for testing standards to assess collision-reducing technologies.

Corresponding author
John P. Swaddle, jpswad@wm.edu

## INTRODUCTION

More than a billion birds die annually in night- and daytime collisions with residential and commercial windows (*Klem, 2014*; *Loss et al., 2014*; *Loss, Will & Marra, 2015*;

*Ocampo-Peñuela et al., 2016*; *Schneider et al., 2018*). It is likely that this pattern of mortality not only creates conservation issues for some avian populations but also raises significant political and socioeconomic barriers to human development of the landscape (*Loss et al., 2014*; *Klem, 2015*). Therefore, there are pressing societal needs to develop technologies that reduce window collisions by birds during daylight hours (*Hager et al., 2013*; *Klem & Saenger, 2013*). Night-time collisions likely occur for other reasons, which we do no address in this study.

It has been proposed that birds collide with windows as they perceive the window as a reflective or transparent surface that extends their environment (*Klem, 2009*). In other words, they do not perceive the window as a solid barrier that they cannot fly through. Though this hypothesis is largely untested in the literature, several companies have produced film treatments that can be applied to the external surface of windows to make the glass more visible to birds despite its reflective properties. Numerous tests of commonly available surface film applications claim that visible vertical stripes of particular widths and separations, as well as some arrangements of dot and shape patterns that do not leave too much open space on the windows, are effective in reducing collisions by birds (*Klem & Saenger, 2013*; *Rössler, Nemeth & Bruckner, 2015*; *Sheppard, 2019*).

Despite the claims in the published literature, we are not aware of any flight tunnel tests that employ lighting conditions that realistically simulate the lighting environments that birds would experience during the day, which may influence the perception of windows as reflective surfaces. A bird that might collide with a window during the daytime will experience daylight on the external surface of the window and artificial light (e.g., at a residential or commercial building) on the internal surface. These lighting conditions will affect whether and how the bird perceives the window as a reflective surface. We expect there would be greatest reflection when there is relatively more light on the external compared with internal surface of a window. Most flight tests of window collisions appear to have been conducted in highly darkened flight tunnels, that will lack important shorter-wavelength light cues, and/or with mirror-reflected light that will alter the polarization of light (*Rössler, Nemeth & Bruckner, 2015*; *Sheppard, 2019*). For those experimental field tests that have used natural-occurring daylight on the external surface of windows (*Klem, 2009*; *Klem & Saenger, 2013*), we are not aware of any experiments that have simultaneously used commonly-occurring artificial light on the internal surface of the windows. Increased brightness of artificial lighting on the internal surface of a window should reduce the perception of reflection. Hence, there is a need to test collision-reducing technologies in daylight with appropriate artificial light placed behind the window.

In addition to considering the external and internal lighting conditions, there has been debate about the relative merits of captive flight tunnel testing vs free-living field testing of factors that influence window collisions (*Klem & Saenger, 2013*; *Sheppard, 2019*). It may be that testing collision-reducing technology in free-living birds is desirable as birds are likely in more naturalistic physiological and behavioral states in such tests. However, field experiments lack as much experimental control as flight tunnel tests and observing collisions in free-living birds is relatively rare and poses welfare and ethical

issues as some of these collisions lead to known damage to experimental subjects (*Klem, 2009*). Flight tunnel tests are also appealing as data can be collected in a relatively short period of time and conditions can be replicated more easily to form a more standardized protocol to guide international efforts in assessing collision-reducing technologies. We attempted to take a middle-ground in this debate by using both domesticated and wild-caught birds in controlled flight tunnel tests, though we recognize it will be important to take steps to test collision-reducing technologies with birds in free-living situations also. The domesticated birds will experience less physiological and behavioral alteration due to handling yet render ecologically-relevant data as their lineage was derived from a free-living species.

Specially, we conducted flight tunnel tests of a new collision reducing technology. BirdShades has produced a window film that is reflective in the shorter-wavelength (e.g., ultraviolet) bands of light and lacks a human-visible striping pattern. The lack of a striping pattern might make the film more esthetically acceptable than the striped films that are currently often used in residential and commercial buildings. To many birds, including both passerines (*Hart, 2001*) and non-passerines (*Goldsmith & Butler, 2005*; *Lind et al., 2013*), a film that reflects shorter wavelengths of light should be visible as a solid surface. To the experimenters' eyes, the BirdShades film had a very light blue tint but otherwise seemed highly translucent. Here, we conducted controlled flight tunnel tests of the efficacy of the BirdShades window treatment in realistic daylight collision conditions.

We performed flight-collision tests with two song bird species, zebra finch (*Taeniopygia guttata*) and brown-headed cowbird (*Molothrus ater*), that are known to be sensitive to shorter-wavelengths of light and hence be good candidate species to benefit from the BirdShades window treatment (*Hart, 2001*; *Aidala et al., 2012*). Specifically, we constructed a long flight tunnel in which the birds flew toward two windows in daylight. Behind each window we placed suitable artificial lights. The windows were tilted back slightly to reflect the sky and, potentially, give the birds the perception of flying into open space. We flew all individuals in three conditions: (i) control, where windows were treated with an external transparent film; (ii) treatment, where the windows were treated with the BirdShades external film; and (iii) both, where one window received the control condition and the other the BirdShades treatment. We placed a fine mist net in front of the windows to prevent actual collisions. We video recorded all flights and analyzed those recordings for the direction and velocity of each bird to assess the likelihood of collision in all experimental conditions.

## MATERIALS AND METHODS

### Experimental subjects and housing

We performed flight trials with two species of songbird (zebra finches $N = 24$, and brown-headed cowbirds $N = 18$) in our custom-built flight tunnel in Williamsburg (VA, USA), to test whether the BirdShades window treatment reduces the risk of collision with windows. When not in the experimental flight tunnel, the zebra finches were housed in an outdoor aviary ($3 \times 3 \times 2.5$ m) and fed ad libitum Volkman science seed mix

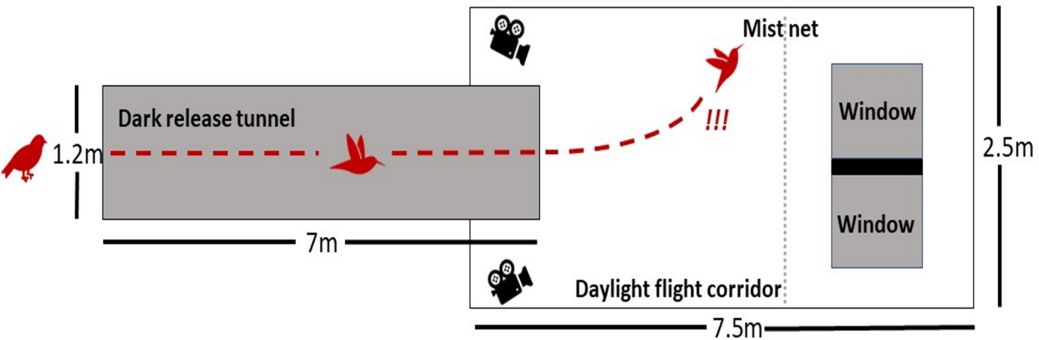

**Figure 1 Schematic of the flight tunnel.** Each bird was released in the dark tunnel (left) and flew to the day-lit flight corridor (right) and collided with a fine mist net before interacting with the two windows. A stlyized flight trajectory is shown in red. The camera icons represent GoPro cameras through which we recorded all flights.

and had access to ad libitum drinking, bathing water, and perches. The zebra finches were selected, somewhat arbitrarily but with the condition that the individuals appeared to fly well in their large free-flight aviaries, from our larger stock colony that we have maintained for 19 years. The brown-headed cowbirds were caught in a baited walk-in trap at a local farm and housed in a long outdoor flight aviary (10 × 3 × 2.5 m) with access to ad libitum food (50:50 blend of commercial chick crumbs with Volkman science seed mix), bathing water, and drinking water. Cowbirds were in these captive conditions for approximately 1 week before flight trials began to assure that all birds were healthy and flying well before the experiment commenced. All zebra finches were banded with a metal leg banded that had a unique identifying number, while all cowbirds were given unique combinations of color bands to identify individuals.

## Flight tunnel and window treatments

The flight tunnel consisted of a long darkened release tunnel (7 × 1.2 × 1.2 m) that opened into a larger day-lit collision tunnel (7.5 × 2.5 × 2.5 m) constructed with fine netting, where the windows were placed (Fig. 1). Birds experienced natural daylight in the collision tunnel and the external surface of the windows experienced natural daylight during all trials (Fig. 2). We conducted all trials between 09:00 and 12:30 from September to December 2019. Around and behind each of the two framed windows, we constructed a lighting box so that the internal surfaces of each window were illuminated with artificial lighting that is representative of residential or commercial buildings. We used two TaoTronics 12 W LED lamps on their highest brightness setting behind each window. Hence all flight tests were conducted in realistic lighting conditions—natural daylight on the external surface and artificial indoor lighting on the internal surface of the windows.

We placed the windows side-by-side with an opaque connector to give the appearance of two windows in a solid wall. There was a 0.5 m gap on both the left and right side of this wall of windows so that birds could divert their flight laterally as to avoid collision. We placed a fine mist net 1 m in front of the windows so that the birds did not actually collide with the windows.

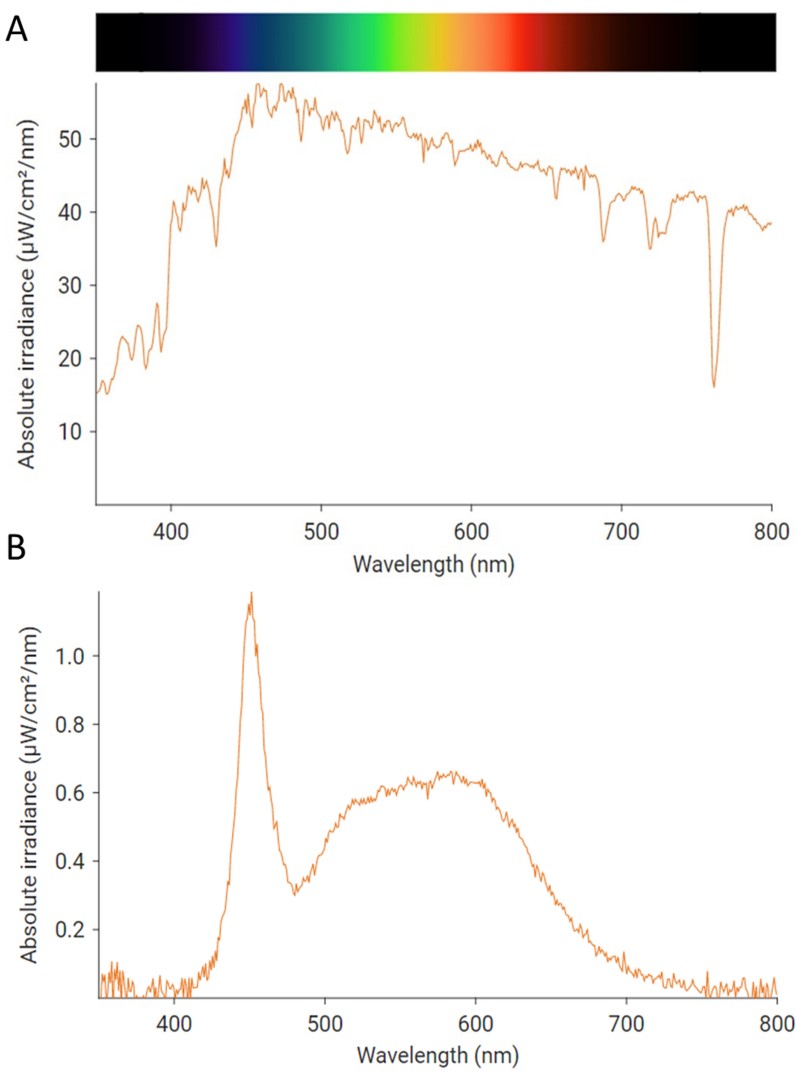

**Figure 2 Example irradiance spectral profile of the light received (A) at the external and (B) the internal surface of windows during flight trials.** The color scheme above the graph indicates human visible colors relative to wavelength of light. Daylight (A) is rich in all visible wavelengths but peaks in the shorter wavelength ranges. The interior lighting (B) also peaked at shorter wavelengths and was far less intense than daylight.

Each bird was exposed to each of three treatment groups, in a balanced order so that the series of presentations and repeated exposure to the flight tunnel did not bias responses by birds. We chose a repeated-measures experimental design as this accounts for among-bird variation in flight behavior that otherwise requires large sample sizes. Birds experienced each treatment group on a separate day, to minimize changes in behavior associated with repeated exposure to the flight tunnel and window. The three treatment groups were as follow: (i) Control. We applied an external film to both windows that had similar physical and spectral-reflectance properties to the BirdShades film except for the reflectance in the ultraviolet wavelength range. This control film was provided by the BirdShades organization. (ii) Treatment. We applied the BirdShades film product to the

external surface of both windows. (iii) Both. In these trials, one window received the control film and the other received the BirdShades film, equally assigned to left and right windows so there was no systematic side bias. This created a forced choice situation for the birds.

Once applied, using instructions supplied by the manufacturer, all window coatings were stored under heavy blankets to ensure that they remained in dark conditions when not in use, so as not to degrade or damage the film coatings. We applied all of the films to Pella 250 Vinyl glass double-glazed replacement windows, as a window commonly found on residential and commercial buildings in the US. The experimental BirdShades film is active in the near UVA range between 300 and 400 nm and is mostly transparent in the human-visible range. Further information about the spectral reflectance and absorption of the window film is available from BirdShades.

Before flight trials and at regular (approximately 30 min) intervals during each session of trials, we measured ambient light irradiance immediately in front of the windows using a handheld spectrometer (WaveGo; Ocean Insight, Largo, FL, USA). This periodic assessment of ambient light allowed us to characterize the general daylight conditions in which the birds experienced the window treatments. We also assessed the light spectra emitted by the artificial lights by holding the spectrometer 0.5 m from the bulb of a single illuminated lamp in an entirely darkened room, to eliminate other sources of light.

## Flight trials and metrics of collision risk

Once the windows were placed in the frames, a flight trial commenced by releasing a single bird from the hand approximately 2 m (from the opening) down the dark release corridor, along with the experimenter emitting a loud startle sound. The bird (in most cases) flew directly away from the experimenter and toward the windows in the open collision tunnel. The windows were 5 m from the opening of the dark release corridor and the mist net was 4 m from the end of the dark release corridor. Hence, for a bird to fly from the release point to the mist net it had to fly for approximately 6 m.

We recorded all flight trials on two GoPro cameras (Hero7 Black at 30 frames per second) so that we could recreate three-dimensional coordinates of birds in flight. In order to maximize flight-path coverage, we used a wide angle (focal length: 15 mm) setting that allowed for coverage of the entire volume of the day-lit collision tunnel. We synchronized the cameras with both acoustic and visual cues. Lens distortion due to the wide-angle modes was corrected for subsequent analysis, using the DWarp Argus package (*Jackson et al., 2016*) implemented in Python 2.7. To recreate three-dimensional flight paths and flight velocity of all birds, we calibrated the cameras and airspace using a wand-based, direct linear transformation technique with sparse-bundle adjustment, implemented in the Argus package in Python 3.6.2 (*Jackson et al., 2016*). Specifically, we digitized the 3D position of the centroid of each bird during the final 15 frames (0.5 s) of each flight and calculated their average velocity across five-frame intervals in this segment (i.e., frames 1–5, 6–10, 11–15 from the end of their flight). We used these three velocity measurements to assess whether birds were altering their flight speed in response to the experimental treatments.

We also visually examined all recorded flights to assess the probability of collision with the windows. We determined that a collision was likely if the bird flew in line with the windows and hit the mist net. If the bird flew too far to the right or left, as to miss the windows, or stopped at least 1 m short of the mist net, we determined that the bird would have avoided the window. We generated our measure of collision/avoidance for each bird in each treatment group. Hence, for each flight of a bird in each treatment group our goal was to produce an assessment of collision risk and a quantification of flight velocity. However, not all birds flew successfully in all trials hence the actual sample sizes in particular analyses were smaller than the total number of birds flown. We report actual sample sizes in association with each statistical analysis.

### Ethics statement

All procedures were approved by William & Mary's Institutional Animal Care and Use Committee (2019-09-22-13861-jpswad). We caught brown-headed cowbirds under US federal bird banding permit 21567 issued to Bryan Watts.

### Statistical analyses

To test whether birds were more likely to "collide" with control or treatment windows when presented with "both" in a forced choice trial, we employed a Wilcoxon matched-pairs signed-ranks test where we scored collisions with a window (control or treatment) as 1 and avoidance as 0, and avoidance of both windows in a trial as 0.5. As this non-parametric repeated-measures test analyzed rank data, the actual numerical values that we chose were not that important. This approach enabled us to examine the within-individual preference for colliding with (or avoiding) control vs treatment windows while not biasing the outcomes with situations where a bird avoided both windows.

To examine the within-individual change in likelihood of window collision for birds in the control compared to the treatment trials we used a Wilcoxon matched-pairs signed-ranks test where we scored collisions with either window as 1 and avoidance of the windows as 0. In this case, the matched-pairs were formed by the same bird experiencing the two treatment conditions.

To test for changes in velocity of birds according to the sequence of the flight video (i.e., frames 1–5, 6–10, 11–15) and the window treatments, we used a linear model in which both time during the flight video and window treatment group were within-subjects fixed factors.

We performed all analyses using IBM SPSS Statistics v25 employing two-tailed tests of probability. All analyzed data are available in the linked dataset.

## RESULTS

### Ambient irradiance and light spectra

Collation of collected irradiance spectra indicated that birds generally experienced the windows in fairly bright daylight (mean ± SEM lux = 23,824 ± 5,278.2; Fig. 2). It is important to note that these spectral measurements are rich in shorter wavelength light, which is typical of daylight conditions. The lamps we used also emitted a peak of shorter
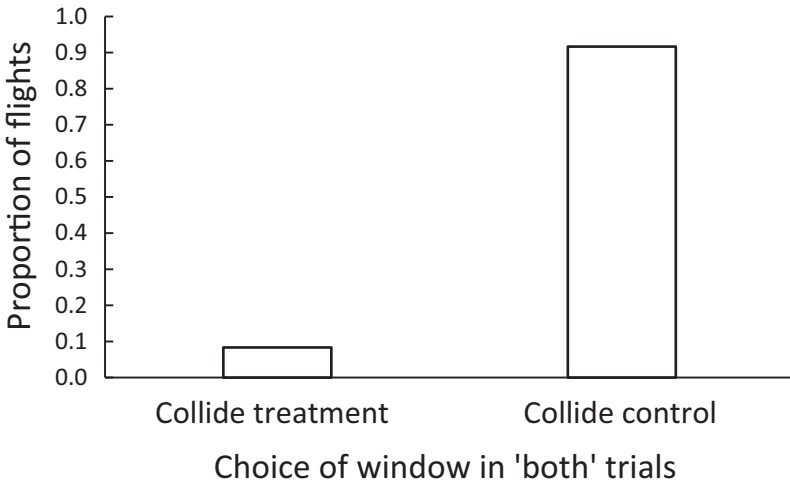

**Figure 3 The proportion of flights in which zebra finches were adjudged to collide with a control or a treatment window, in forced-choice "both" trials.** For zebra finches, there was a 91% reduction in collision risk when windows were treated with the BirdShades product compared with the control product.                                                                       

wavelength light with another broad peak at intermediate to longer wavelengths of the visible spectrum (lux = 422; Fig. 2). From our scans of residential and office lighting in our area, this spectrum and brightness is typical of the internal surface of many windows (mean = 334, SD = 329, minimum = 12, maximum = 1,847; all measurements in lux).

## Zebra finches

Among the zebra finches, the birds were more likely to collide with a control window than a treatment window when given the choice between the two in a forced choice (both) trial. In 60% ($N$ = 12 of 20) of these "both" choice trials, birds were judged to "collide" with one of the windows. Of these 12 collision events, 8.3% (1 of 12) were with the BirdShades-treated window and 91.7% (11 of 12) were with the control-treated window (Fig. 3). These differences could be interpreted as a 91% reduction in collision risk when windows are treated with the BirdShades product compared with the control product (Wilcoxon $Z$ = 2.63, $N$ = 20 pairs, $p$ = 0.0085; Fig. 3).

However, although such tests are popular in the published literature and are useful as a point of comparison to other published studies, these forced choice "both" trials are not realistic of how birds experience windows in nature. In real situations, all windows would be treated in the same manner and it is more relevant to examine the differences in the probability of collision when both windows are either experimentally-treated or control-treated (i.e., likelihood of collision in control vs treatment trials, for each bird).

When both windows were control-treated, the zebra finches were less likely to avoid the windows (15%, 3 of 20 trials) than to collide with the windows (85%, 17 of 20 trials) (left pair of bars on Fig. 4). However, when the windows were both treated with the BirdShades film, the zebra finches were marginally more likely to avoid the windows (57.9%, 11 of 19 trials) than collide with the windows (42.1%, 8 of 19 trials) (right pair of bars on Fig. 4). Taken together, the zebra finches were approximately 50% less likely to

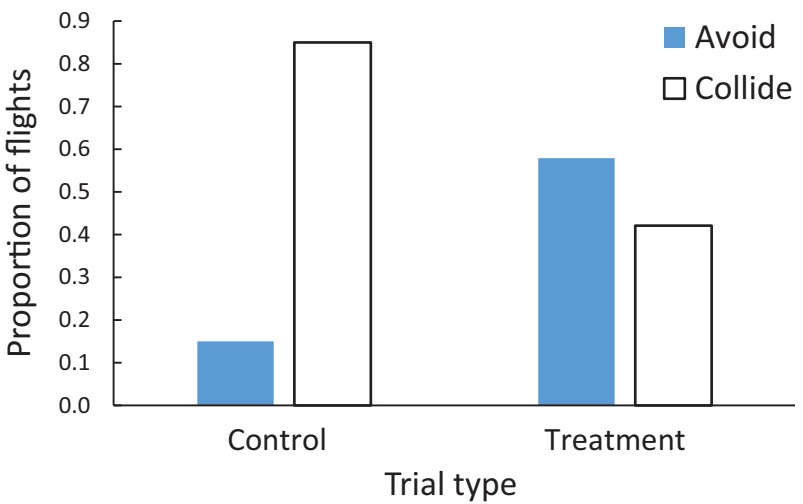

**Figure 4 Proportion of flights when zebra finches were adjudged to collide with windows (open bars) or avoid windows (filled bars) in either the control or treatment trials.** The zebra finches were approximately 50% less likely to collide with a window if the windows were treated with the BirdShades film compared with the control film.

collide with a window if the windows were treated with the BirdShades film compared with the control film (Wilcoxon, $Z = 3.29$, $N = 17$ pairs, $p = 0.001$; Fig. 4).

Zebra finches adjusted their flight velocity according to the window treatments ($F_{2,26} = 3.78$, $p = 0.036$) and decelerated as they approached the windows in all treatments ($F_{2,26} = 90.46$, $p < 0.0001$; Fig. 5). Specifically, the birds flew approximately 25% slower when the windows were treated with the BirdShades film (planned contrast of control vs. treatment, $F_{1,13} = 11.20$, $p = 0.005$). The forced choice "both" condition, where one window was control-treated and the other was BirdShades-treated, rendered intermediate values that were not different to the control trials (planned contrast of control vs both, $F_{1,13} = 0.778$, $p = 0.394$). As all groups decelerated at approximately equal rates, it appears that birds adjusted their velocity early in their flight.

## Brown-headed cowbirds

We observed qualitatively similar responses to the window treatments among the brown-headed cowbirds as the zebra finches. The cowbirds were more likely to collide with a control window than a BirdShades-treated window when given the choice between the two in a forced choice trial (i.e., "both" trials). In 88.2% ($N = 15$ of 17) of these forced choice "both" trials, cowbirds were judged to likely collide with either window. Of these collision events, 20% (3 of 15) were with the BirdShades-treated window and 80% (12 of 15) were with the control-treated window (Wilcoxon, $Z = 2.02$, $N = 17$ pairs, $p = 0.043$; Fig. 6). This could be interpreted as a 75% reduction in the risk of collision.

When both windows were control-treated, the cowbirds all collided with the windows (100% of 17 trials) with no avoidance (left bar on Fig. 7). However, when both windows in a trial were the BirdShades-treated windows, the cowbirds avoided the windows in 29.4% (5 of 17) of trials and were adjudged to collide with the windows in

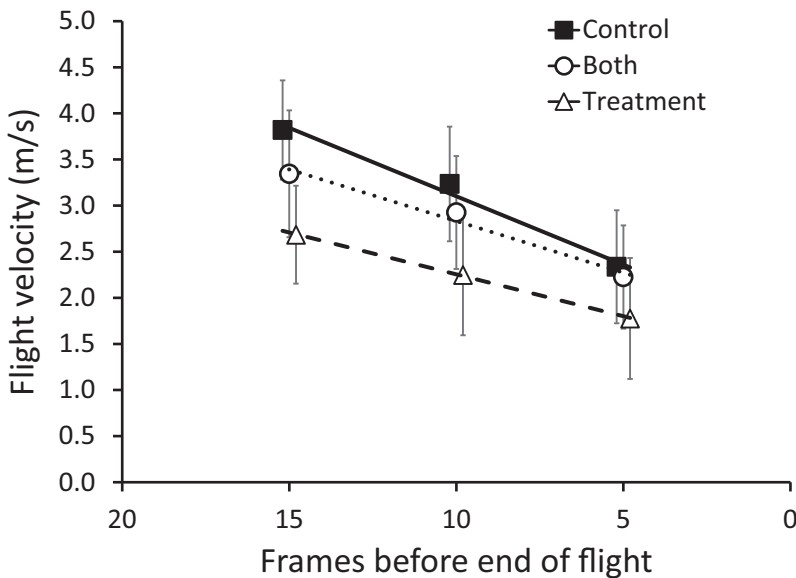

**Figure 5 Mean (±95% CI) flight velocity across sequential five frames of flight (15–11, 10–6, 5–1, counting from the end of flight) for zebra finches.** Treatment groups are as follows, control (filled squares), treatment (hollow triangles), and both (hollow circles) trials. Lines connecting points are least-squares lines of best fit. Zebra finches flew approximately 25% slower when the windows were treated with the BirdShades film compared with the control condition.

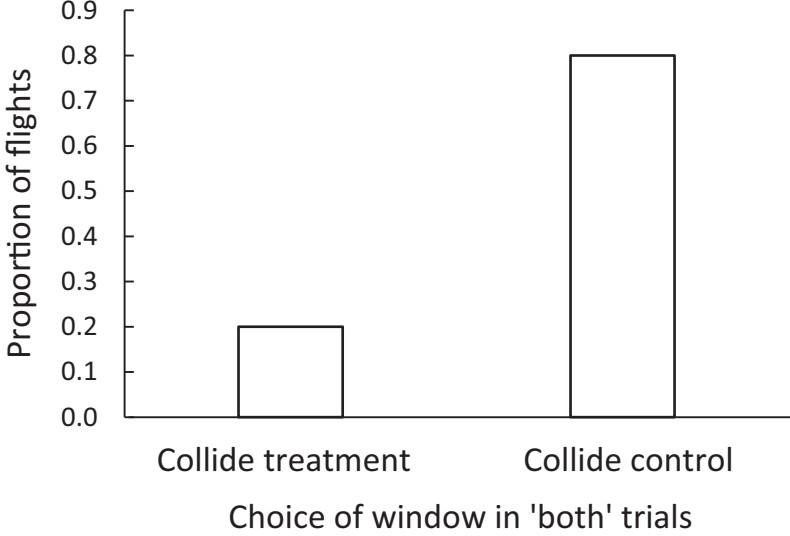

**Figure 6 The proportion of flights in which brown-headed cowbirds were adjudged to collide with a control or a treatment window, in forced-choice "both" trials.** Brown-headed cowbirds experienced a 75% reduction in the risk of collision under the BirdShades compared with the control treatment.

70.6% (12 of 17) of these flight trials (right pair of bars on Fig. 7). Taken together, the cowbirds were at least 30% less likely to collide with a window if the windows were treated with the BirdShades product compared with the control product ($Z = 3.29$, $N = 17$ pairs,
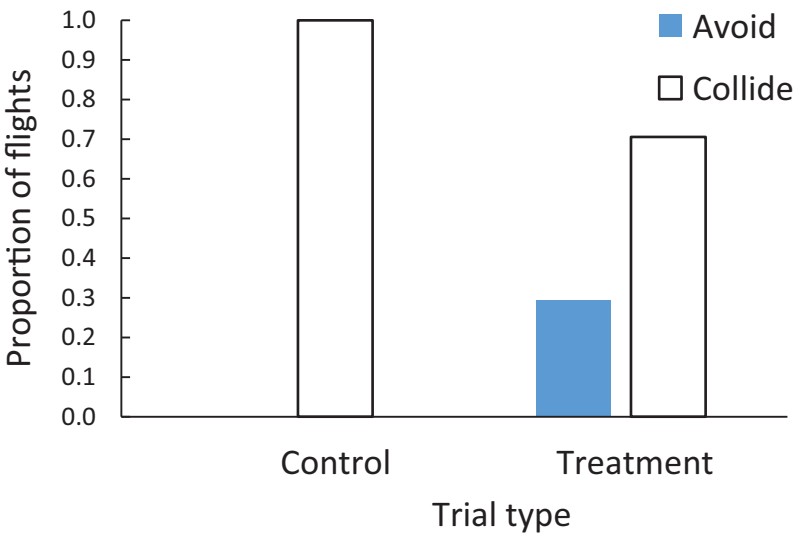

**Figure 7 Proportion of flights when brown-headed cowbirds were adjudged to collide with windows (open bars) or avoid windows (filled bars) in either the control or treatment trials.** Brown-headed cowbirds were at least 30% less likely to collide with a window if the windows were treated with the BirdShades product compared with the control product.

$p = 0.001$; Fig. 7). However, that number could be larger as we did not observe any avoidance in the control trials. It could be argued that the BirdShades treatment improved avoidance by a very large magnitude compared to zero avoidance in the control treatment.

Due to digitization issues with some calibration files, the sample size of videos we could analyze for flight velocity of cowbirds was reduced. There were few birds where we had a measurement of flight velocity in all three treatments. Hence, we decided to adopt an alternative linear model approach with the time sequence during a video as a repeated-measures factor but treatment as an among-subjects factor due to lack of replication of the same bird across all treatments. This analysis indicated that, as with the zebra finches, the cowbirds decelerated as they approached the windows ($F_{2,62} = 40.09$, $p < 0.001$) and that there was a non-statistically supported trend for the cowbirds to fly approximately 25% slower when presented with a BirdShades treated window ($F_{2,31} = 2.59$, $p = 0.091$; Fig. 8). As with the zebra finches, the cowbirds adjusted their velocity early in their flight and decelerated at similar rates across treatments (Fig. 8).

## DISCUSSION

Our data indicate that zebra finches and brown-headed cowbirds are more likely to avoid a harmful collision with a window if the window is externally treated with the BirdShades film. As we compared collision risk to a control treatment that also received a window film but that was not reflective in the shorter (ultraviolet) wavelengths of light, we can further conclude that increased surface reflection in the ultraviolet ends of the avian-visible spectrum assists with avoiding window collisions. This conclusion does not preclude that other "colors" of reflected light might also be important. As many avian species are sensitive to ultraviolet light, but especially small passerines (*Hart, 2001*;

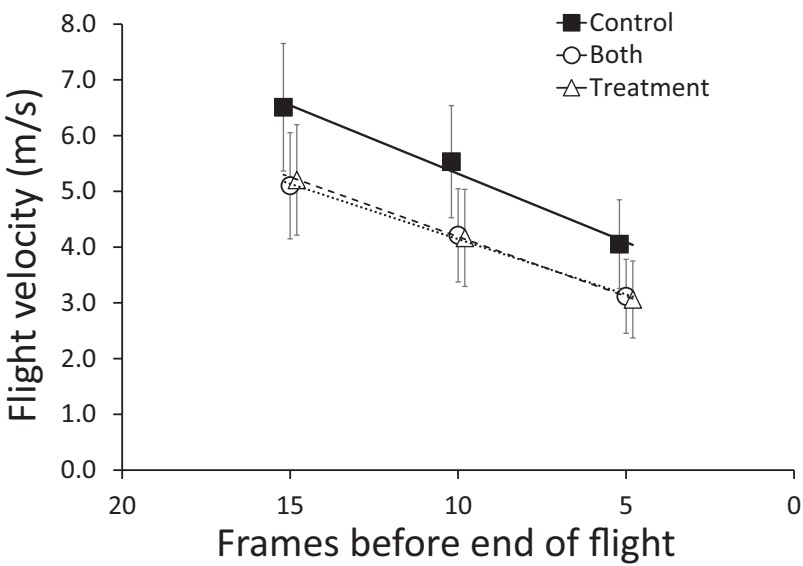

**Figure 8 Mean (±95% CI) flight velocity across sequential five frames of flight (15–11, 10–6, 5–1, counting from the end of flight) for brown-headed cowbirds.** Treatment groups are as follows, control (filled squares), treatment (hollow triangles), and both (hollow circles) trials. Lines connecting points are least-squares lines of best fit. There was a non-statistically supported trend for the brown-headed cowbirds to fly approximately 25% slower when presented with a BirdShades treated window compared with controls.

*Aidala et al., 2012*), this observation indicates that the BirdShades film could help protect birds from window collisions in many localities.

Not only was the overall risk of collision reduced for both zebra finches and brown-headed cowbirds when the BirdShades film coating was applied to the windows, but the velocity of flights was also reduced by approximately 25% during the last 0.5 s of flight. As the birds were close to the windows during this video sequence, the reduction in flight velocity is most parsimoniously explained as an effect of the window treatments. This may mean that when a collision occurs in nature it will be with less force and, therefore, less likely to cause irreparable damage to the bird. Obviously, we did not allow the birds to actually strike the windows in our flight tunnel, but the deceleration we observed indicates a protective value of window treatments that has not been quantified in other studies.

In part of our study, we presented birds with an industry-standard forced choice test (i.e., our "both" treatment) where the control and treatment windows were presented side-by-side in the same flight trial (cf. *Rössler, Nemeth & Bruckner, 2015*; *Sheppard, 2019*). In separate flight trials we presented the birds with situations where all windows in a trial were either control or BirdShades-treated. We argue that the latter type of comparison, where all of the windows in a trial have been treated with the same film, is more ecologically salient as it will be common that adjacent windows on a building will be treated in the same manner (*Hager et al., 2013*). It would be a rare situation where birds would make an instantaneous choice between windows that have been treated differently and those windows are presented side-by-side, as in the "both" treatment. Of note, the

effect size in the more ecologically-relevant comparison of flight in the "control" vs "treatment" trials is consistently smaller (approximately 50% in zebra finches and 30% in brown-headed cowbirds) than that estimated in the forced choice situation of the "both" trials (approximately 90% in zebra finches and 75% in brown-headed cowbirds). These observations suggest that many previously published estimates of effect sizes, that have used forced choice paradigms (*Rössler, Nemeth & Bruckner, 2015*; *Sheppard, 2019*), might also be over-estimating the efficacy of collision-reduction technologies. Therefore, we urge caution in applying data from forced choice trials in predicting any effects in changing the rate of collisions with windows in common in-situ conditions. If we take the effect size we have isolated in these more ecologically-relevant trials at face value, a reduction in bird collision rate of 30–50% would still offer substantial protective value and potentially avoid millions of bird deaths annually.

As far as we are aware, our experiment may be the first to document the effectiveness of any external window treatment on the risks of window collision when the birds are flying in lighting conditions that realistically replicate those of a day-lit building window with indoor lighting. The sensory ecology of birds will be greatly affected by lighting conditions and influence the ways in which the birds perceive and attempt to avoid collision risks (*Martin, 2011*). Specifically, we ensured that the external surface of the windows received ambient daylight, which is rich in shorter-wavelength light at all times of this study, while also illuminating the internal surface of windows with representative artificial light. The interplay of the external and internal lighting conditions with the reflective and absorptive properties of the window (plus film and glue applied) will affect whether the birds perceive the window as a reflection, a solid structure, or otherwise. We contend that many other flight tunnel tests that have not exposed birds to realistic lighting conditions, most notably those that fly the birds in dark tunnels without daylight on the external surface (i.e., birds' side) of windows (*Rössler, Nemeth & Bruckner, 2015*; *Sheppard, 2019*), have limited application to reducing window collisions in daylight conditions. Their results might apply to interpreting how birds respond to lit windows during night flights, depending on whether the internal surface of those windows has been illuminated with suitable artificial lighting.

Other than illustrating the apparent effectiveness of the BirdShades film product, we propose this study raises methodological issues that can contribute to the development of window collisions testing standards. We advocate that scientists and industry collaborates to develop standardized methodology for testing and comparing the effectiveness of collision-reducing technology. Further, we highlight the following conditions, which are currently not commonplace in window collision testing. First, if flight tunnels are used, they must attempt to simulate more naturalistic ecological conditions for birds, including realistic internal and external lighting conditions and they should avoid forced-choice window trials but instead present trials in which all windows are treated the same way in a no-choice design. Birds will rarely experience a forced-choice situation in nature and our data indicate that forced-choice measurements likely overestimate the effectiveness of technologies.

Methodological design of window-collision tests must account for other additional factors. In constructing treatment groups in flight tunnel tests, we advocate for the use of positive control trials in which the windows are manipulated but not treated with the specific film coating that will be tested. This allows the experimenter to understand the influence of the general presence of film and glue independently of the precise reflective and absorptive properties of the test film. In this study we used a transparent film as a positive control. Negative controls, where windows are entirely untreated cannot deconfound other associated factors (e.g., presence of film and adhesive, or general alteration of the surface of glass). If only one control treatment is possible, we advocate for the use of positive controls. Further, when exposing birds to the experimental treatments, we advocate for the inclusion of repeated-measures designs where possible. By exposing the same individuals to all treatments it is possible to minimize the effects of among-individual variance in behavior, motivation, and physiological state on the metrics of collisions. Repeated-measures approaches have proven useful in many other contexts (*Shaffer, 1979*; *Swaddle, 1997*; *Swaddle & Biewener, 2000*; *Boulton et al., 2015*) but have yet to be applied to the collisions literature. Importantly, this allows the experimenter to employ smaller sample sizes, potentially reducing the cost and time involved in adequately testing products. By making window collisions testing more affordable and efficient we might gain more information about the efficacy of products. It is possible to employ repeated-measures designs in field trials, if birds are individually color-banded and those birds are resident within the area.

In addition to the aforementioned points, again applied to flight tunnel tests, we should continue to compare the performance of recently wild-caught individuals to performance of captive-reared birds. This will help us to understand the influence of prior housing and stress levels on metrics of collisions. Finally, where possible, we recommend that experimenters assess the velocity of flight and/or the force of collisions. In our study we showed that flight velocity can be lowered even when collisions are still likely to occur. By gaining more information about the impacts of any potential collision we can be more precise in predictions about how birds will be affected.

Avian collisions with windows are an issue of importance to global conservation as perhaps up to a billion birds die each year when they fly into closed windows (*Klem, 2014*; *Loss et al., 2014*; *Loss, Will & Marra, 2015*; *Ocampo-Peñuela et al., 2016*; *Schneider et al., 2018*) and the problem is geographically widespread (*Basilio, Moreno & Piratelli, 2020*). Though external window treatments are generally affordable they can be expensive to apply if not affixed to windows at the time of building construction. The high cost is often associated with getting access to a high-story external window surface. If the films can be made durable, they are promising solutions if planned for with sufficient forethought. In North America, there have been recent calls for and some legislative action to require bird-friendly glass on buildings (e.g., 2020 legislation in New York City, USA) but it is unclear how this will be achieved, especially as the tinted glass can be more expensive than other options. Externally-applied products, such as the one we tested from BirdShades, could play an affordable role in solutions but only if we test the products in ways that are realistic with how birds interact with windows in nature. Hence, there is

pressing need to evaluate, using the criteria we describe above, externally-applied window films and glass that is tinted in particular spectral ranges in order to provide affordable, sustainable, and effective conservation solutions.

Overall, we advocate for the continued use of flight tunnel tests as they offer controlled conditions that are more easily replicated by different research groups. Such flight tunnel tests need to be adapted to deliver realistic lighting conditions (on both external and internal surfaces of windows) and not use forced choice experimental designs. Further they should employ appropriate positive controls and, when possible, use repeated-measures approaches so that studies can further isolate the effects of any window treatments. However, these flight tunnel tests should be accompanied with field testing that meets similar rigorous experimental standards but place the windows in the actual context that birds experience—on real buildings. We have yet to perform field tests with the BirdShades product.

## CONCLUSIONS

Overall, our data support a conclusion that the BirdShades external window film reduces the risk of collision by zebra finches and brown-headed cowbirds by approximately 30–50%. It could be claimed that these estimates should be higher, according to the outcomes of the forced choice trials (perhaps 80–90%), but we interpret the forced choice design to lack ecological relevance and over-estimate what is more likely to happen in the field. This is a deficit of the flight tunnel testing window-collisions literature in general. Furthermore, when collision does occur, it is likely that the birds will strike the windows with less force as the BirdShades film treatment results in 25% reduction in flight velocity close to the point of impact. Therefore, we propose that the BirdShades product could be largely effective in mitigating and preventing window collisions, particularly for the large number of avian species who are sensitive to shorter wavelengths of light. It is important that this flight tunnel testing is supported by installations on buildings so that the actual effects on avian mortality can be documented. Further, we advocate for researchers and industry to collaborate in devising standards for assessing window collisions, and start this discussion by suggesting several important features that are not yet commonly incorporated into flight tunnel testing.

## ACKNOWLEDGEMENTS

We thank Sally Mullis and Lyra Swaddle for assistance with flight tunnel construction and data collection. We thank Bryan Watts for assistance with permissions to catch wild birds. We also thank Brandon Jackson for guidance with 3D flight digitization.

### Funding

This work was supported by a Plumeri Fellowship from William & Mary to John Swaddle and by BirdShades Innovations GmbH. The funders had no role in study design, data collection and analysis, decision to publish, or preparation of the manuscript.

## Grant Disclosures

The following grant information was disclosed by the authors:
William & Mary.
BirdShades Innovations GmbH.

## Competing Interests

BirdShades Innovations GmbH provided partial funding of this project and provided the window films that were used in the experiments. BirdShades had no influence on the collection, analysis, interpretation, or presentation of the data and did not contribute to the preparation of this manuscript.

## Author Contributions

- John P. Swaddle conceived and designed the experiments, performed the experiments, analyzed the data, prepared figures and/or tables, authored or reviewed drafts of the paper, and approved the final draft.
- Lauren C. Emerson performed the experiments, analyzed the data, authored or reviewed drafts of the paper, and approved the final draft.
- Robin G. Thady performed the experiments, analyzed the data, authored or reviewed drafts of the paper, and approved the final draft.
- Timothy J. Boycott performed the experiments, analyzed the data, authored or reviewed drafts of the paper, and approved the final draft.

## Animal Ethics

The following information was supplied relating to ethical approvals (i.e., approving body and any reference numbers):

This work was approved by the Institutional Animal Care and Use Committee of William & Mary (2019-09-22-13861-jpswad).

## Data Availability

Raw data is available as a Supplemental File.

## Supplemental Information

Supplemental information for this article can be found online at http://dx.doi.org/10.7717/peerj.9926#supplemental-information.

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
