# Peer review of "Ultraviolet-reflective film applied to windows reduces the likelihood of collisions for two species of songbird"

_PeerJ, doi:10.7717/peerj.9926_

## Round 0.1 · original submission · Major Revisions

As you can see, you have three reviewers who have made some excellent suggestions. Please address these thoroughly and let me see your revised manuscript and detailed responses to these comments.

Reviewer 1 ·

Basic reporting

53 I don´t think that many companies have produced film treatments. There is a number of companies that produced those films, not so many.
56 Vertical stripes are not the only pattern that is recommended, also dots and any other pattern. It is relevant not to leave open spaces larger that 2-4 inches.
75 It is not only important to consider the artificial light conditions inside the room, but also the fact that, some buildings present windows that look like tunnels. In those cases, the transparence of the glass should also be considered.
94 It is worth to mention the fact that not all species of birds are able to see ultraviolet patterns. Perhaps the authors can specify groups of birds that can see ultraviolet patterns (Most Passeriformes can, as far as I know), but non Passeriformes can´t see them.
108 Was the ceiling of the tunnel made of net or, how where the windows reflecting the sky? If the ceiling of the tunnel was made of net, is there a chance that birds change the direction of their flight toward the sky instead of going to the glass windows?
106-116 The authors describe the methods on the introduction. I think this section should not be included with so much details in the introduction.

Experimental design

135-137 As I mention before, I wonder if the open area covered with fine netting changed the direction of the birds which may try to fly up instead going to the windows.
120-123. The objective of the article at the end of the Introduction instead at the beginning of Methods.
127 The colony of birds have been living a reproducing in captivity during 19 years? If that is correct, I suspect they may be used to be in closed areas. There is not need for them to fly a lot.
128 How long the cowbirds were kept in captivity before the tests?

Validity of the findings

The article present introduced changes to flight tunnels in order to have more "realistic" conditions. The use of other species of birds should be used without using a method to prevent collisions in order to proof birds are flying toward the glass window and not flight to the open area covered with net.

On the discussion section, I suggest to include more references to other similar or releated studies to discuss your findings. For example, other studies mention the advantages and disadvantages of flight tunnels?
Reduction of risk of collisions in 30-50% is high enough to considere the product effective to reduce bird window collisions?

·

Basic reporting

Clear and well done. Some relevant missing information that other researchers would consider essential, about the quantitative description of the treatment tested.

Experimental design

Credible and acceptable. As valid as other similar protocols used to address the same subject.

Validity of the findings

Data collection, analysis and interpretation are accurate, justifiable, reasonable, and judged valid.

Additional comments

The paper is well written, clear, and effectively presented. The figures are relevant and useful.

Authors provide minimal information on the BirdShade film. Other researchers could not test, replicate, or further validate this product using similar or other experimental protocols without knowing how to inquire about obtaining it. There is minimal, little to no quantitative description of the UV signal (reflection component) of the BirdShade film. Previous studies in the literature interpret strength of UV-reflection and UV-absorption as determining factors in avian signal perception guiding avoidance of window treatment deterrence. Recommend adding more quantitative description of UV signal used to deter window strikes, or explain why this information was not relevant and included.

The use of internal artificial light is novel, but its importance and relevance to the deceptive illusion of reflections for birds, although well-known, is reasonable yet uncertain given its variation in interior settings. That the interior light was standardized and modeled after existing windows provides an ability to measure its influence. This a valid novel component to the experimental protocol, and advances our collective knowledge.

Overall, the study reports important information from a modified tunnel-testing protocol addressing assessment, product, and discussion about bird-window collision deterrence in a growing literature on the subject having extraordinary importance for avian conservation.

·

Basic reporting

The basic reporting of results is good, but I have made some comments on the figures and the way results are presented.
The organization of the writing can be improved significantly in the introduction and discussion. It could be made clearer and could maybe benefit from subtitles of topics.

You need more luterature references in both your introduction and discussion. I have pointed out special places where these are missing in the attached and annotated PDF

Experimental design

The experimental design is appropriate for the testing done in this study.

Validity of the findings

These are important findings for urban conservation. The authors test a brand-new UV coated film that has the potential to prevent window collision and decrease the speed in which birds fly into windows. Importantly, this film does not have a printed pattern and is instead coated with UV, which makes it almost invisible to humans and thus more appealing for architects to use.
It would be important to test this film on real buildings. I suggest the authors make this a stronger recommendation at the end.

Additional comments

Introduction
The introduction has all the right elements, but it would benefit from a reorganizing. I suggest you write some subtitles (which you can later delete) and develop each idea separately.
Importantly, the last paragraph is all methods. You should move that to the next session and instead finish with a paragraph stating your objectives, short methods, and what you expect (your hypotheses).

Methods
These are generally well written and clear.
Results
I made some comments on the figures. The legends need to be stand-alone and complete. They should be able to explain the figure without having to read the paper. I also suggest you standardize the aesthetics of your figures (colors, font, font sizes). I suggested deleting two figures that don’t really show anything additional.
I suggest re-reading your results and looking for places where things are not clear. I got a little lost and often had to go back to the beginning of the paragraph. Use your figures wisely and avoid repeating the figure information in the text, instead refer to the figure.
Discussion
As the introduction, this section could use a lot of work. I suggest using topic subtitles and sticking to a subject for each paragraph. Organize your ideas and present them in order.
You are missing a lot of citations that would better support your claims. I would like to see a second version with many more references to other experiments, even if not on bird-window collisions.
You don’t discuss the implications for bird conservation this experiment could have. I suggest you spend a paragraph discussing why these experiments are important and what is the potential for this film to reduce collisions. Also, is this a cheap solution? Is it easy to apply? Does it have to changed every certain time? Discuss what would be required for this film to succeed and for people to actually use it. Put your experiment in the context of this issue and discuss its potential!

---

## Round 0.2 · accepted · Accept

I have read your responses to the reviewers and find that you've done a credible job of answering them. You haven't always done what they asked — but you've made it clear when you haven't and why you haven't. In particular, I believe you have addressed the comments of the reviewer who requested "major revisions."